



# Effect of surficial geology mapping scale on modelled ground ice in Canadian Shield terrain

H. Brendan O'Neill[1], Stephen A. Wolfe[1], Caroline Duchesne[1], Ryan J. H. Parker[1]
[1]Geological Survey of Canada, Natural Resources Canada, 601 Booth St., Ottawa, ON, Canada

5    *Correspondence to*: H. Brendan O'Neill (hughbrendan.oneill@nrcan-rncan.gc.ca)



**Abstract.** Ground ice maps at circumpolar or hemispherical scales offer generalised depictions of abundance across broad geographic regions. In this paper, the effect of surficial geology mapping scale on modelled ground ice abundance is examined in the Slave Geological Province of the Canadian Shield, a region where the geological and glacial legacy has produced a landscape with significant variation in surface cover. Existing model routines from the Ground ice map of Canada (GIMC) were used with a 1:125 000 scale regional surficial geology compilation and compared to the national outputs, which are based on surficial geology at 1:5 000 000 scale. Overall, the regional scale modelling predicts much more ground ice than the GIMC due to greater representation of unconsolidated sediments in the region. Improved modelling accuracy is indicated by comparison of outputs to available empirical datasets due to improved representation of the inherent regional heterogeneity in surficial geology. The results demonstrate that the GIMC significantly underestimates the abundance and distribution of ground ice over Canadian Shield terrain. In areas with limited information on ground ice, regional-scale modelling may provide useful reconnaissance-level information to help guide field-based investigations required for planning infrastructure development. The use of current small-scale ground ice mapping in risk or cost assessments relating to permafrost thaw may significantly influence accuracy of outputs in areas like the Canadian Shield where surficial materials range from bedrock to frost-susceptible deposits over relatively short distances.

**Copyright information**

# 1 Introduction

Ground ice is a critical component of permafrost terrain and provides geotechnical strength to frozen ground. However, climate change is causing permafrost thaw and ground ice melt (Smith et al., 2022), resulting in widespread terrain subsidence (O'Neill et al., 2023), hillslope failure (Lewkowicz and Way, 2019), changes to hydrologic conditions (Walvoord and Kurylyk, 2016), and damage to infrastructure (Doré et al., 2016).

In this paper, the influence of the scale of surficial geology input data on modelled ground ice abundance is examined in an assessment of data uncertainty (e.g., Riseborough et al., 2008). The study region includes the Yellowknife-Grays Bay transportation corridor, a proposed infrastructure route that would service mineral-rich areas in central Canada. The surficial geology of the region is heterogeneous, ranging from bedrock or thin till veneer cover in uplands to thicker deposits of fine-grained till or marine sediments. Significant variation in the surficial geology occurs over relatively short distances compared to the thicker and more continuous drift that covers the Interior Platform geological province to the west (Miall, 2015). The modelling methodology was developed by O'Neill et al. (2019) for the production of the Ground ice map of Canada (GIMC; O'Neill et al., 2022). It uses an expert-system approach implemented in a Geographic Information System (GIS) to predict the relative abundance of excess ice in the top 5 m of permafrost from relict (buried glacial), segregated, and wedge ice. Assessments of the GIMC outputs have indicated that the national-scale surficial materials dataset underrepresents





unconsolidated sediments and heterogeneity in surficial materials, and thus ground ice abundance, in some areas of the
Canadian Shield due to the scale of the mapping (Kokelj et al., 2023; O'Neill et al., 2019; Wolfe et al., 2021). Subedi et al.
(2020) demonstrated the underestimation of relict ice abundance near Lac de Gras, where glacial ice was interpreted from
coring, but where no relict ice is modelled on the GIMC due the lack of mapped thicker till units. Though broad-scale products
may poorly represent empirical evidence of ice-rich permafrost (Kokelj et al., 2023), ground ice information from small-scale
mapping such as the International Permafrost Association (IPA) Circum-Arctic Map of Permafrost and Ground-Ice Conditions
(IPA map; Brown et al., 2002) and the GIMC are used in generalized assessments of infrastructure risks and costs related to
permafrost thaw (e.g., Clark et al., 2022; Hjort et al., 2018; Streletskiy et al., 2023), as more detailed ground ice information
is not available in a standardized digital form for Canada or worldwide.

The objectives of this paper are to compare ground ice model outputs generated using more detailed surficial geology
mapping and those from the national scale GIMC. Specifically, we 1) describe differences between national (1:5 000 000) and
regional (1:125 000) scale surficial compilations over a large region of the Canadian Shield spanning varied surficial geology,
2) highlight and quantify resulting discrepancies in modelled ground ice abundance, 3) validate model outputs with available
information on ground ice in the region, and discuss implications on the accuracy of the representation of ground ice on the
Canadian Shield, and 4) compare differences in modelled ground ice along a proposed transportation infrastructure route with
the IPA map and GIMC, and discuss the implications for risk and cost assessments.

## 2 Study area

The study area extends from Great Slave Lake near Yellowknife, NT, northward to the coast of Coronation Gulf, NU (Fig. 1),
representing a >600 km latitudinal transect in Canadian Shield terrain with varying surficial geology. The region was shaped
by Late Wisconsin glaciation and includes gently undulating terrain to moderately rugged topography, with numerous bedrock
outcrops (Dredge et al., 1999). Deglaciation near the coast occurred from about 10 500 BP to 9 600 BP, whereas southwest of
Contwoyto Lake, deglaciation and vegetation establishment occurred by about 8 500 BP (Dredge et al., 1999). Till deposits
and bedrock dominate the surficial geology. Tills are stony diamicton; those derived from sedimentary and metasedimentary
rocks include an appreciable silt-clay fraction, whereas tills sourced from granitic and gneissic rocks have a more sandy matrix
(Dredge et al., 1999). Glacial Lake McConnell deposited fine-grained lacustrine sediments in the Great Slave Lowlands prior
to its recession (Wolfe et al., 2014). Near the Coronation Gulf Coast, emergence of the landscape following deglaciation left
fine-grained marine deposits up to elevations of about 200 m asl (Dredge et al., 1999).

The climate at Yellowknife is cold and continental with a mean annual air temperature (1981-2010) of -4.3 °C, and
at Kugluktuk, about 120 km west of the north end of the study area, -10.3 °C (Canadian Climate Normals, 2020). Vegetation
across this climate gradient transitions from boreal forest near Yellowknife to grassland-lichen-moss tundra in the north
(Latifovic, 2019). Permafrost is discontinuous near Yellowknife (Figure 1), and occurs in peatlands and areas with ice-rich,





unconsolidated sediments; annual mean ground temperatures are -1.4 to 0 °C (Morse et al., 2016). Annual mean ground

temperatures near Lac de Gras are about -6 °C, and -5 to -7 °C on the coastal plain east of Coronation Gulf (Wolfe et al., 2017).

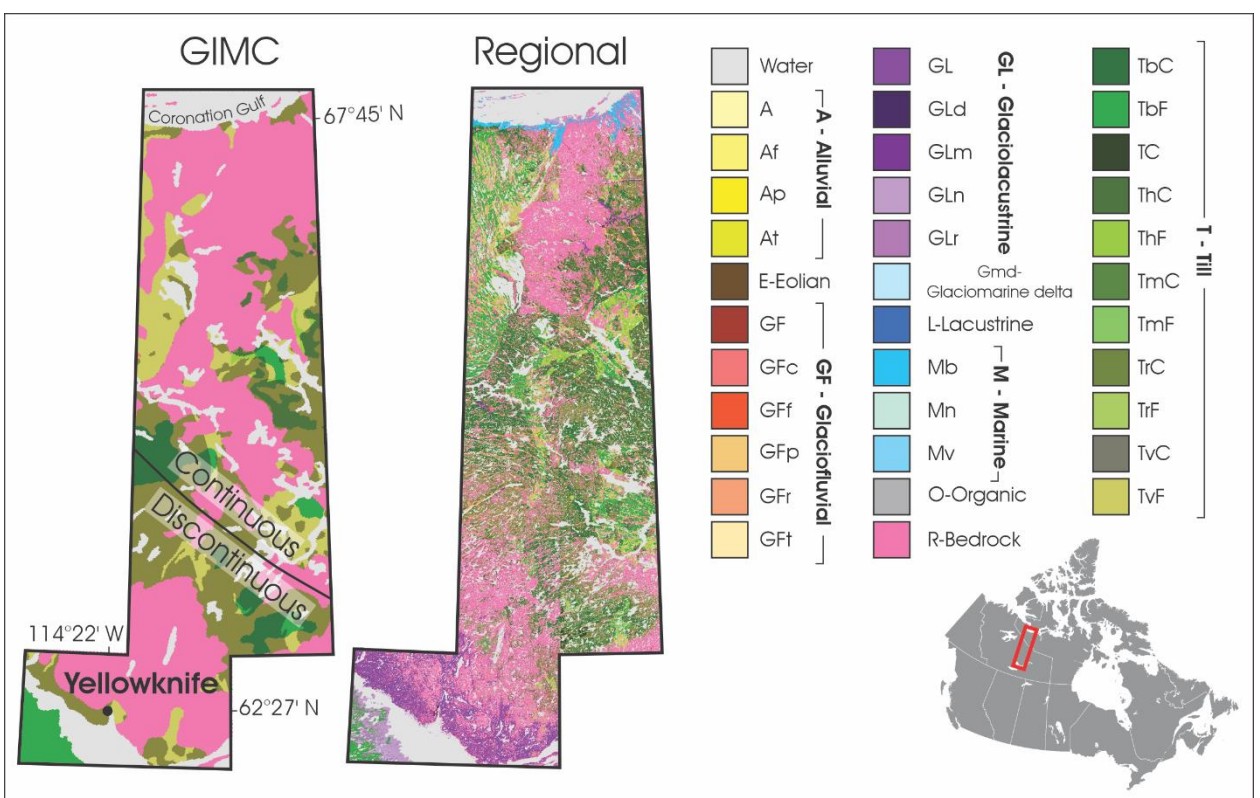

**Figure 1. Surficial materials classes from the compilation used for the GIMC (O'Neill et al., 2019) and regional compilation. Surficial**
**geology unit abbreviations are from the Geological Survey of Canada Surficial Data Model (Deblonde et al., 2019, Appendix 2).**
**Suffixes: f = fan, p = plain, t = terrace(d), c = ice contact, r = ridged (T)/esker (GF)/beach (GL), d = deltaic, m = moraine, b = blanket,**
**n = nearshore, v = veneer, h = hummocky. Till suffixes 'C' and 'F' indicate coarse- and fine-grained units, respectively, based on**
**underlying bedrock geology (see Sect. 3).**

## 3 Methods

Eleven 1:125 000 scale surficial geology maps (hereafter called the regional compilation, RC) were used for the regional

ground ice modelling (Geological Survey of Canada, 2017a, b, 2016a, b, 2015, 2014b, a; Kerr, 2018, 2014; Olthof et al., 2014;

Stevens et al., 2017). As with the 1:5 000 000 scale national surficial compilation, till units were classified into fine- and

coarse-grained dominated textures based on underlying bedrock geology (Dredge et al., 1999; O'Neill et al., 2019).

Ten Canadian Geoscience Maps (CGMs) conformed to the GSC's Surficial Data Model, ensuring standardised

legends for the surficial geology units (Deblonde et al., 2019). Olthof et al. (2014) was produced using predictive methods

based on Landsat imagery and did not conform to the data model as it included an "Undifferentiated" sediment unit. This unit

occurs only along shorelines and represents spectral mixing between the land surface and water bodies within Landsat pixels.



These pixels were limited in coverage and excluded from the modelling. Major townsites (e.g., Yellowknife) or existing highway routes mapped as anthropogenic deposits were also excluded. Four bedrock unit types were combined into a single bedrock class. This resulted in 34 surficial material classes on the RC (excluding water), compared to a total of 29 for all of Canada on the GIMC.

Surficial units that appeared on the national-scale mapping retained their model parameter values from the GIMC. Units that were not represented at the national scale were assigned parameters based on a review of surficial geology-ground ice associations informed by the map unit legends and observations from prior investigations (e.g., Dredge et al., 1999; Kerr et al., 1996; Subedi et al., 2020; Wolfe, 1998; Wolfe et al., 2017).

The vector shapefiles for all surficial geology map sheets were merged in Esri™ ArcGIS™ and rasterized with a pixel size of 250 m, whereas pixels were 1000 m on the GIMC. The pixel size was chosen to preserve small surficial geology polygons and detail around complex shorelines of small lakes, though underlying map units for the surficial geology datasets are commonly larger and map units from the other model data layers are highly generalized. Therefore, modelled abundance in an individual pixel represents the average condition of the broader mapping unit in which it occurs.

Differences between the regional and national surficial compilations, and the modelled ground ice outputs (relict, segregated, wedge) were examined for the whole study region using summary coverage statistics produced from the raster layers. A combined ground ice abundance output layer was produced from the relict, segregated, and wedge ice layers following the method used for the GIMC (O'Neill et al., 2022). The discrepancies in combined ice abundance were examined by producing a difference map between the regional modelling and GIMC. The GIMC raster was resampled to 250 m pixel size and a raster calculator was used to subtract the GIMC combined ice values from those of the regional model output. Differences between the regional and GIMC outputs were also considered in 33 areas of interest (AOIs) measuring 15 x 15 km to examine the influence of surficial geology complexity on modelled outputs at both scales. Three AOIs were selected manually to capture a range of conditions in surficial geology heterogeneity, and 30 additional AOIs were selected randomly within the study region. The influence of heterogeneity in surficial geology on resulting modelled ground ice was explored using relations between the number of surficial units mapped in each AOI, and the total number of ground ice classes represented, which has a theoretical maximum of 15 (3 ground ice types with 5 possible abundance classes for each). The average combined ice abundance was also calculated for the 33 AOIs using the coded numeric values for each ice abundance class (0 to 5 for none to very high) and the proportion of each area underlain by each abundance class. This enabled comparison of the regional and GIMC combined ice abundance in each AOI.

## 3.1 Validation

Published ancillary data was used to assess the model results. For relict ice this included limited borehole information where glacial ice was specifically interpreted in the studies (Subedi et al., 2020; Wolfe, 1998; Wolfe et al., 1997). Segregated ice conditions were assessed in the Great Slave Lowlands using a dataset of mapped lithalsas – ice-cored ridges or small hills formed by segregated ice accumulation in mineral soils (Stevens et al., 2012), and observations of landform-surficial geology



associations from other studies (Kerr et al., 1996; Dredge et al., 1999; Morse et al., 2023). The wedge ice model results were analysed using point observations of ice-wedge polygon networks from five of the Canadian Geoscience Maps. These observations were not available from all map sheets as their inclusion is dependent on whether the original surficial mappers recorded these periglacial features. Therefore, the wedge ice validation is not systematic but provides a general idea of

associations between ice wedge polygons and mapped surficial geology conditions over nearly half of the study region. The underlying surficial geology class and modelled ground ice abundance from both the GIMC and the regional modelling were extracted for each point observation in the validation datasets to gain qualitative insight into the accuracy of modelled abundance classes.

### 3.2 Infrastructure corridor assessment

The distribution of modelled ground ice was examined along the route of the proposed Yellowknife-Grays Bay transportation infrastructure corridor and compared with the GIMC and IPA map to gauge the utility of the different products in assessments of thaw sensitivity. The polyline representing the corridor from Northwest Territories to the Nunavut coast extends about 750 km, and represents a portion of the Tibbitt to Contwoyto Winter Road route, and proposed all-season roads extending to Grays Bay (e.g., Morse et al., 2023; Figure 2). Modelled ice abundance underlying the route was summarized using the Tabulate

Intersection function in ArcGIS Pro, which sums the total length of the line within each ice abundance class, derived from vector polygon conversions of the regional modelling and GIMC raster outputs and the IPA map shapefile (Brown et al., 2002). Ice abundance classes were compared along the corridor between the three mapping products. Though the legends differ between the regional modelling and the GIMC (excess ice in top 5 m of permafrost), and IPA map (visible ice in top 10-20 m), equivalent categories were compared (e.g., low vs. low, medium vs. medium). The average combined ice abundance along

the entire corridor was also calculated using the values for each ice abundance class (0 to 5 for none to very high) and the proportions of the route underlain by each abundance class, which allows for a rudimentary aggregate comparison between the three models.

## 4 Results

### 4.1 Surficial materials

The RC includes 34 surficial material classes, many more than the 8 represented on the national compilation within the study area (Fig. 1). Sediments on the RC include units of till (n=11), bedrock (n=1), glaciolacustrine, lacustrine, and glaciomarine (n=7), alluvial (n=4), glaciofluvial (n=6), marine (n=3), eolian (n=1), and organic material (n=1). The national compilation includes units of till (n=6), bedrock (n=1), and glaciofluvial material (n=1).

        Unconsolidated sediments and organic terrain associated with ground ice are mapped over more of the study area on

the RC (47%) than the national compilation (39%; Table 1). This is despite much greater water coverage on the RC (28%) than the national mapping (14%). Bedrock occupies far less of the study area on the RC (26%) than on the national compilation





(47%), as do till veneers (17% vs. 30%), which contain little to no ground ice. Thicker till units that are associated with ground ice are far more prevalent on the RC (19% vs. 9%; Table 1). Furthermore, glaciolacustrine, glaciomarine, lacustrine, and marine units cover 8% of the study region on the RC, but these potentially ice-rich units are not represented at all at the national

scale.

Table 1. Summary of surficial material types and the percent of the study area that they cover on the national compilation for the GIMC and the regional compilation (RC).

| Surficial Material type(s) | GIMC % study area | RC % study area |
| --- | --- | --- |
| Bedrock | 47 | 26 |
|  |  |  |
| *Till veneers* | 30 | 17 |
| *Tills (less till veneer)* | 9 | 19 |
| *Glaciofluvial* | <1 | 2 |
| *Glaciolacustrine, glaciomarine, lacustrine, and marine* | N/A | 6 |
| *Organic* | N/A | 1 |
| *Eolian + Alluvial* | N/A | <1 |
| Sum unconsolidated & organic | 39 | 47 |
|  |  |  |
| Water | 14 | 28 |

## 4.2 Modelled ground ice abundance


The greater distribution of unconsolidated sediments on the RC results in markedly greater modelled ground ice abundance over this Canadian Shield region than on the GIMC (Fig. 2). Among all ice types, the area of the study region with no ground ice is significantly less on the regional modelling than on the GIMC (-33 to -58% difference between RC and GIMC; Fig. 3), which is mainly due to the greater representation of bedrock at the national scale. There is also less area (-39% difference)

modelled with negligible ice abundance, mainly due to the lower coverage of till veneers on the RC compared to the GIMC (Fig. 3; Table 1). The overrepresentation of bedrock and till veneer is due to less detailed mapping on the national compilation, and the photographic reduction of surficial geology units during the conversion of legacy maps, originally at 1:1 000 000 scale, to the 1:5 000 000 scale representation in Fulton (1995), which forms the basis for the GIMC compilation. In producing the surficial geology map of Canada (Fulton, 1995), units smaller than a minimum size visible at 1:5 000 000 scale for a printed

wall map were removed. On the Canadian Shield, this resulted in the disappearance of pockets of thicker unconsolidated sediment units where till veneer and bedrock were the dominant material. As a result, the regional modelling includes greater area (+21 – +200% difference) represented by low, medium, high, and very high combined ice abundance (Fig. 3).

The regional model output has far more area with low (+167% difference), medium (+104%), and high (+70%) relict ice abundance than the GIMC, due to the greater area mapped as thicker glacial deposits that could host preserved glacial ice





(Figs. 3,4). For segregated ice, more area with low and medium abundance (+22 and +147% difference; Fig. 3) is present on the regional model output due to the increased representation of various frost-susceptible deposits (Figs. 3,5). High modelled segregated ice abundance occurs in marine sediments near the Coronation Gulf coast, but no areas were modelled with high abundance on the GIMC as these deposits are not represented (Fig. 6). Similarly, a greater area with low (+98% difference) and medium (+189%) wedge ice abundance is present on the regional mapping (Figs. 3,6). There is no modelled high wedge

ice abundance in either of the model outputs because the length of time is insufficient for the modelled accumulation since deglaciation and/or emergence.

Areas with notably higher modelled ice abundance on the RC than the GIMC include north of Great Slave Lake, where glaciolacustrine deposits are widely represented on the regional compilation (Figs. 1, 2). Near Napaktulik Lake and northward toward the Coronation Gulf, the abundance of all ice types is higher than on the GIMC predominantly due to more

widespread mapping of thicker till units.

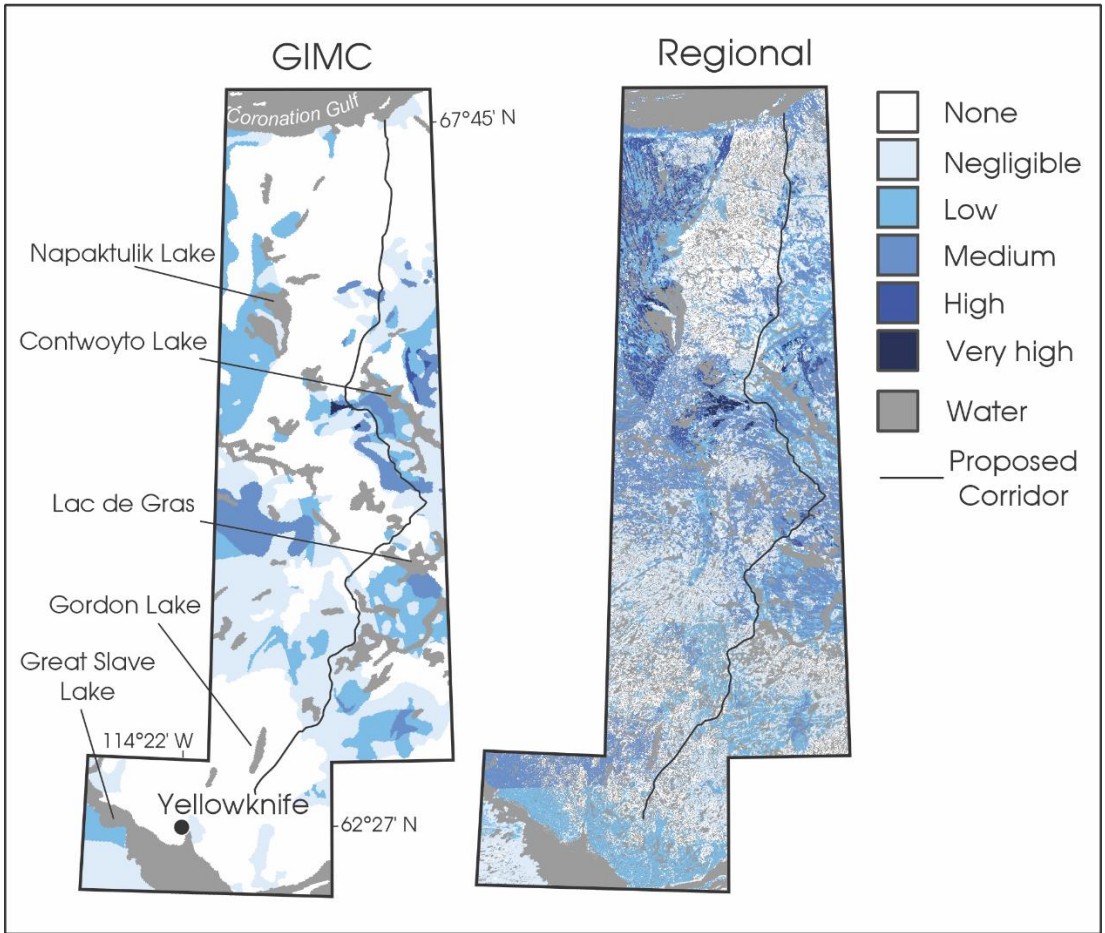

**Figure 2. Modelled combined ice abundance (relict, segregated, wedge ice) outputs for the study area from the GIMC (left; O'Neill et al., 2022) and regional modelling (right). The proposed transportation infrastructure corridor is marked by the black line (modified from Morse et al., 2023).**






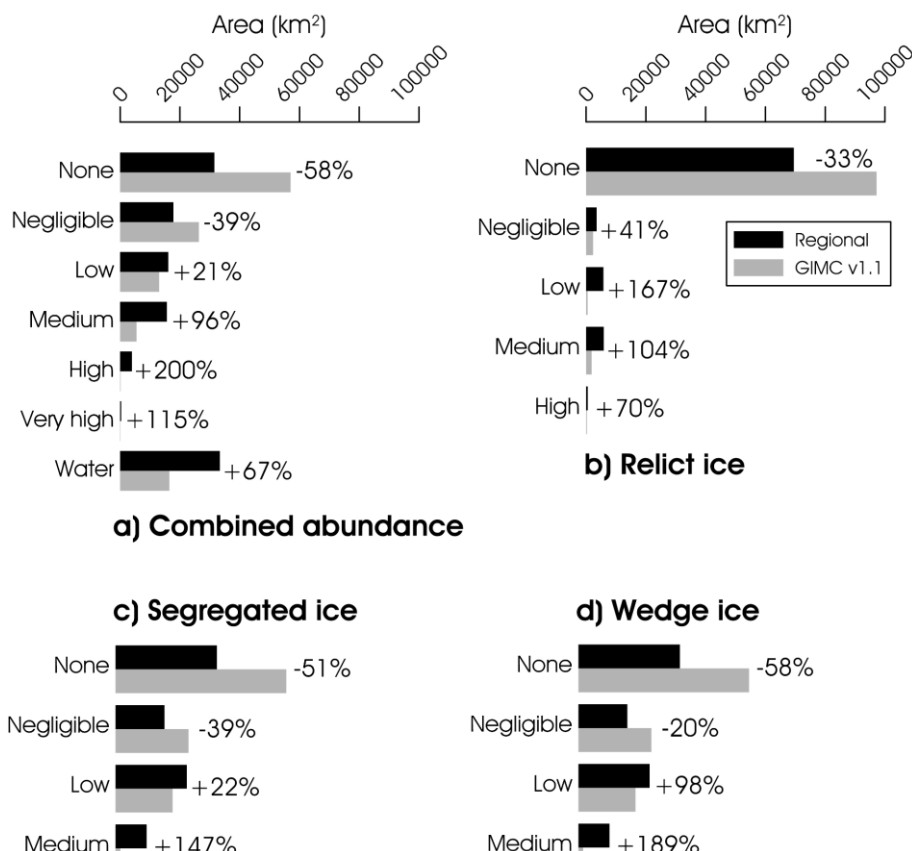

**Figure 3. Bar charts depicting the area of the study region occupied by each ground ice abundance class, and water, for modelling based on the regional compilation, and that of the GIMC (O'Neill et al., 2022). The numbers above the bars indicate the percent difference in the areas depicted by regional-scale modelling vs. the GIMC. Percent difference is calculated as: (Regional-GIMC)/((Regional+GIMC)/2) x 100. Negative values indicate less area represented on the regional modelling than on the GIMC.**


Overall, the study-area wide differences in outputs reveal higher ground ice abundance at the regional scale than on the GIMC. However, the heterogeneity of surficial materials at finer spatial scales creates variation in these differences across the region (Fig. 4a). Typically, fewer surficial classes are represented on the GIMC in the 15 x 15 km areas of interest (AOIs), which is associated with fewer ground ice classes being represented (Fig. 4b). This commonly results in lower average combined ice

content in AOIs on the GIMC, as indicated by most points lying well above the 1:1 line in Fig. 4c. As is expected with respect to mapping scale, high discrepancies in average combined ice content (Fig. 4c) occur in AOIs with greater heterogeneity of





surficial cover and where variation occurs over short horizontal distances (Fig. 4d,e; Subedi et al., 2020). In contrast, where surficial cover is more homogeneous, differences in modelled ice abundance due to mapping scale are negligible (Fig. 4f, g);

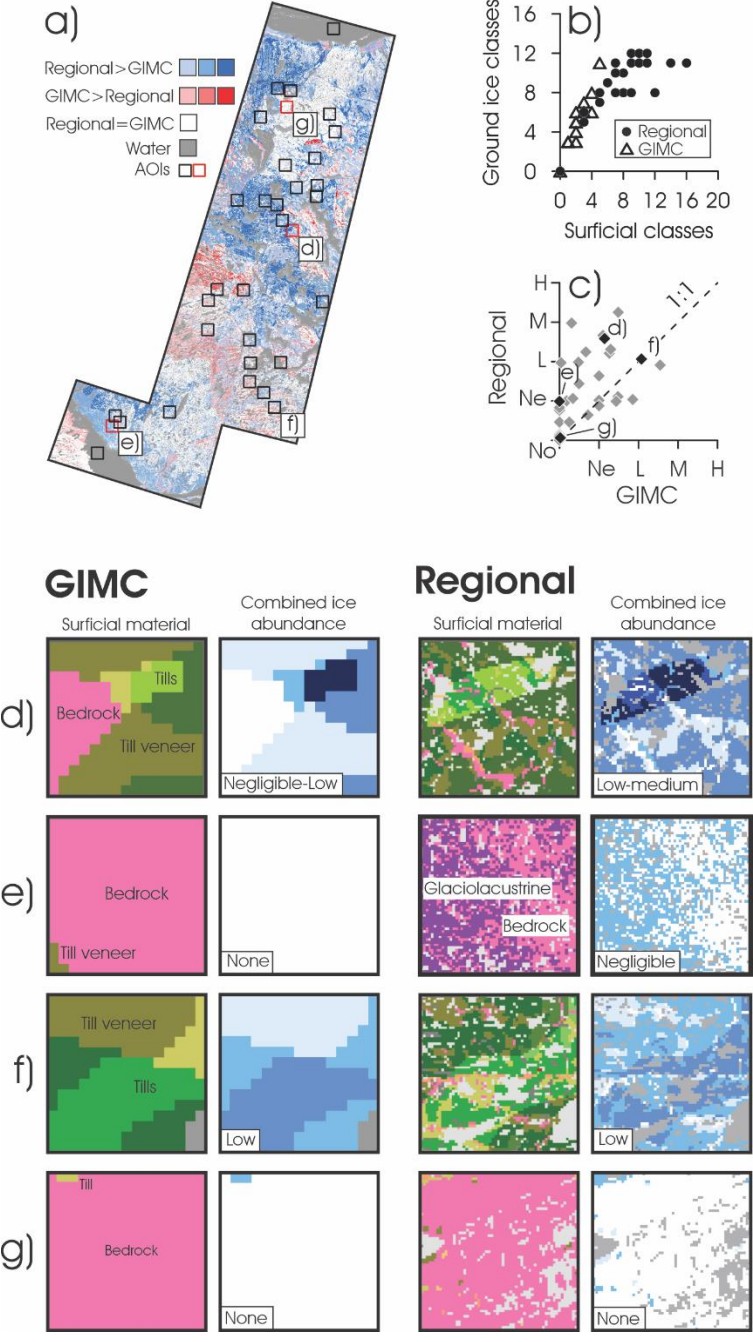

**Figure 4. Difference map between a) modelled regional and GIMC combined ice abundance; greater differences are indicated by darker shades; squares indicate random (black) and selected (red) 15 x 15 km areas of interest (AOIs) used in the analysis, b) relation between the number of surficial materials classes and number of ground ice classes represented in each AOI shown in a), c) average ground ice content in each AOI for the GIMC vs. regional outputs; No, Ne, L, M, H = none, negligible, low, medium, and high,**





**respectively, d) to g) examples showing surficial materials and modelled average combined ice abundance from the four AOIs labelled in a) for the GIMC and regional modelling outputs. Note: only dominant surficial materials within each AOI are labelled by text in d) to g), see Fig. 1 for complete legend.**

these AOIs lie along the 1:1 line in Fig. 4c. These results have broader implications for interpreting the accuracy of the GIMC in different regions. For example, a lower discrepancy between GIMC and regional modelling is likely in many areas of the western Arctic, where thicker, frost-susceptible deposits are relatively homogeneous compared to deposits in this study region.

Similarly, little discrepancy would be expected in other areas of the Canadian Shield where bedrock is dominant at both mapping scales (Fig. 4g).

## 5 Discussion

### 5.1 Validation

The representation of ground ice is improved in several areas on the regional mapping where observations are available. The
presence of relict ice has been interpreted from boreholes drilled in upland tills north of Lac de Gras (Subedi et al., 2020). The GIMC indicates no relict ice in this area, but the regional modelling includes appreciable areas with low and medium abundance, and smaller areas of high abundance in the vicinity, due to the improved representation of thicker glacial deposits (Fig. 5a-c). Relict ice has also been observed in glaciolacustrine delta and esker deposits in the region (Wolfe, 1998; Wolfe et al., 1997). These materials are better represented on the RC, which likely contributes to overall improved accuracy in the
region (Table 1). However, specific locations and conditions associated with field observations indicate potential for improvements in the modelled relict ice output. First, the sampled esker from Wolfe et al. (1997) occurs in an area mapped as till veneer on the RC (Fig. 5d), resulting in no modelled relict ice. The potential for relict ice in these linear features, which may not be represented in the surficial geology mapping polygons from Canadian Geoscience Maps unless they are sufficiently large, could be better represented by incorporating GIS line features from CGMs in the modelling (Fig. 5d). Second, modelled
relict ice is absent at two drilled sites where preserved ice was found in glaciolacustrine deltaic sediments (Wolfe, 1998), because this area was inundated during deglaciation, and the GIMC model routine assumes that relict ice melts if the land was inundated. This assumption is largely valid at the national scale where small ice-marginal features are not mapped. However, this ice would have been buried and preserved during the progradation of the delta (Wolfe, 1998), which postdates the inundation that causes the ice to melt in the model. Therefore, the model rule concerning the melt of relict ice by inundation
could be improved for this regional-scale modelling by creating exceptions for some ice-marginal deposits.





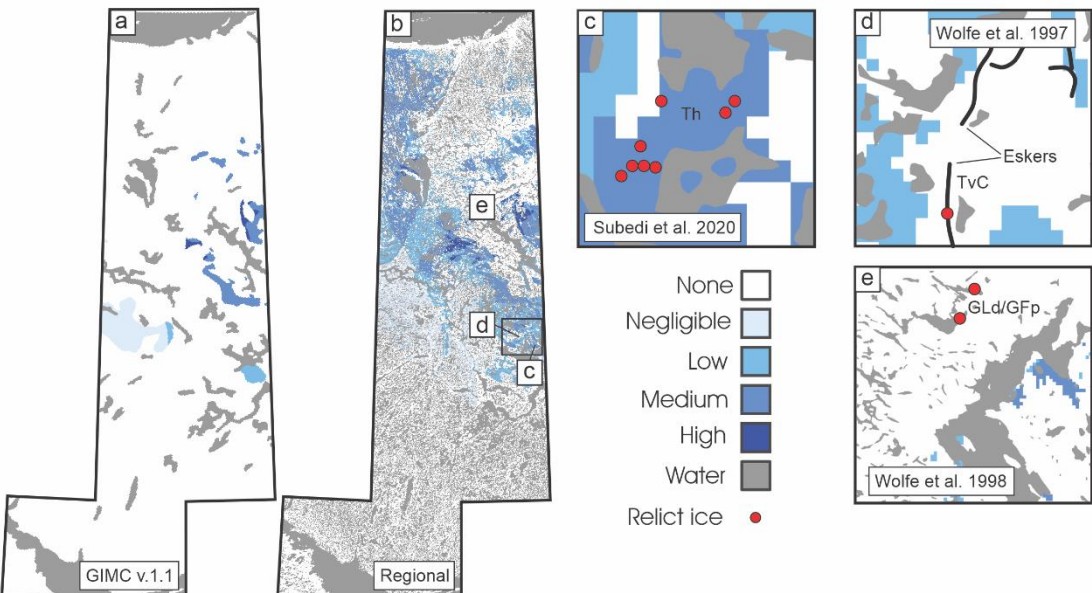

**Figure 5. Modelled relict (glacial ice) abundance from a) GIMC v.1.1, b) the regional modelling, with labels showing areas with interpreted relict ice from field investigations: c) borehole locations in hummocky till (Th) uplands (Subedi et al., 2020), d) borehole drilled in an esker (Wolfe et al., 1997), in an area mapped as coarse-grained till veneer (TvC) on the RC, with esker line features from the Lac de Gras surficial map added (Geological Survey of Canada, 2014b), and e) boreholes drilled in sediments interpreted as a glaciolacustrine delta (GLd) in the field (Wolfe, 1998), and mapped as a glaciofluvial plain (GFp) on the RC.**

Based on available information, the accuracy of the regional segregated ice output is improved compared to the GIMC. Thousands of lithalsas and lithalsa-lake complexes are found in glaciolacustrine sediments within the former limit of Glacial Lake McConnell in the Great Slave Lowlands (Kokelj et al., 2023; Wolfe et al., 2014). The regional modelling includes these glaciolacustrine deposits, such that most mapped lithalsas fall in areas with low to medium segregated ice abundance, whereas the GIMC portrays the area as mostly devoid of segregated ice, with small areas of negligible abundance (Fig. 6a-d). Near the Coronation Gulf coast, high segregated ice abundance is modelled in marine deposits using the regional compilation where retrogressive-thaw slumps are observed (Dredge et al., 1999; Morse et al., 2023); segregated ice is mainly absent in these areas on the GIMC (Fig. 6). Mudboils and solifluction lobes are common on tills derived from dolomite and argillite north of Napaktulik Lake (Dredge et al., 1999), suggesting the presence of frost-susceptible sediments in settings where the regional modelling predicts low to medium segregated ice abundance, but where the GIMC largely predicts no segregated ice (Fig. 6e).





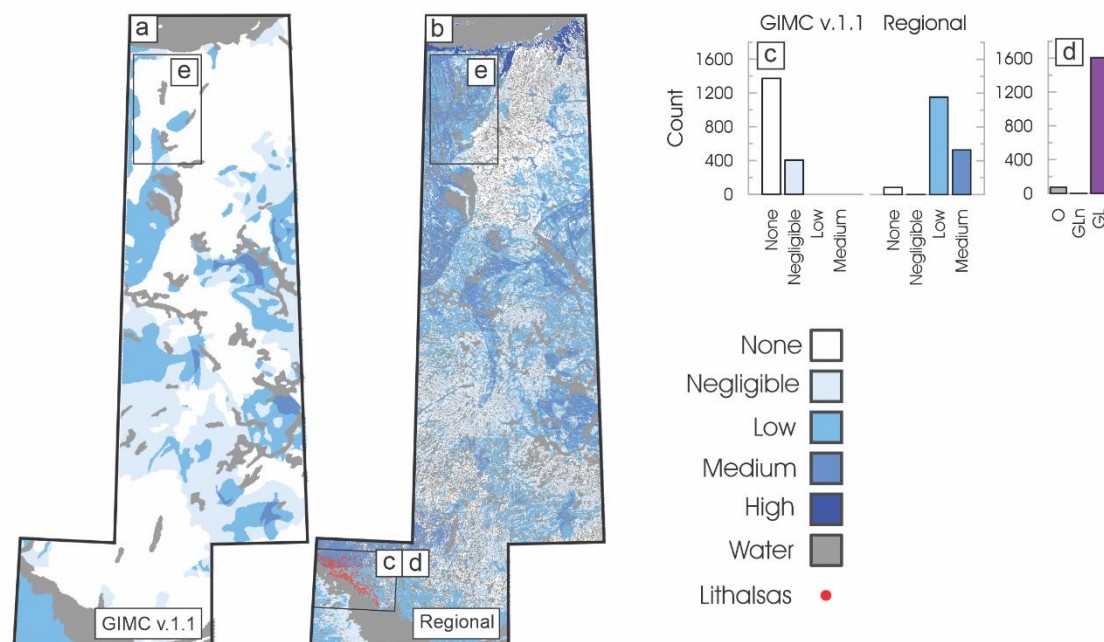

**Figure 6. Modelled segregated ice abundance from a) GIMC v.1.1, b) the regional modelling, with inset (labelled c,d) showing mapped lithalsas within the modelled domain (n=1683) in the Great Slave Lowlands from Stevens et al. (2012), c) histogram of mapped lithalsas from b) and corresponding ground ice abundance on the national and regional-scale mapping, respectively, d) histogram of surficial geology from the regional compilation underlying each mapped lithalsa point, and e) region with widespread tills mapped on the RC, where mudboils and solifluction lobes are common, indicating frost-susceptible sediments (Dredge et al., 1999).**

Validation of relative wedge ice abundance is challenging because volumetric estimates from the region are lacking. However, point observations of ice-wedge polygons from the CGM mapping demonstrate improved accuracy of the regional modelling, with the bulk of polygons occurring in areas with low or medium wedge ice abundance (Fig. 7a-c). In contrast, most of these observations occurred in areas with either no or negligible modelled abundance on the GIMC (Fig. 7c). Surficial materials on the RC associated with the highest densities of mapped polygons include organic, alluvial, marine/glaciomarine, glaciolacustrine, and glaciofluvial deposits (Fig. 7d,e). This is consistent with independent findings from mapping based on satellite imagery along the length of the proposed transportation infrastructure corridor (Morse et al., 2023), and field observations of well-developed ice wedge polygons in areas with thicker peat (Karunaratne, 2011; Subedi et al., 2020). In these units, there is reasonable agreement between areas with modelled low or medium abundance and mapped areas of ice wedge polygons (Fig. 7f).



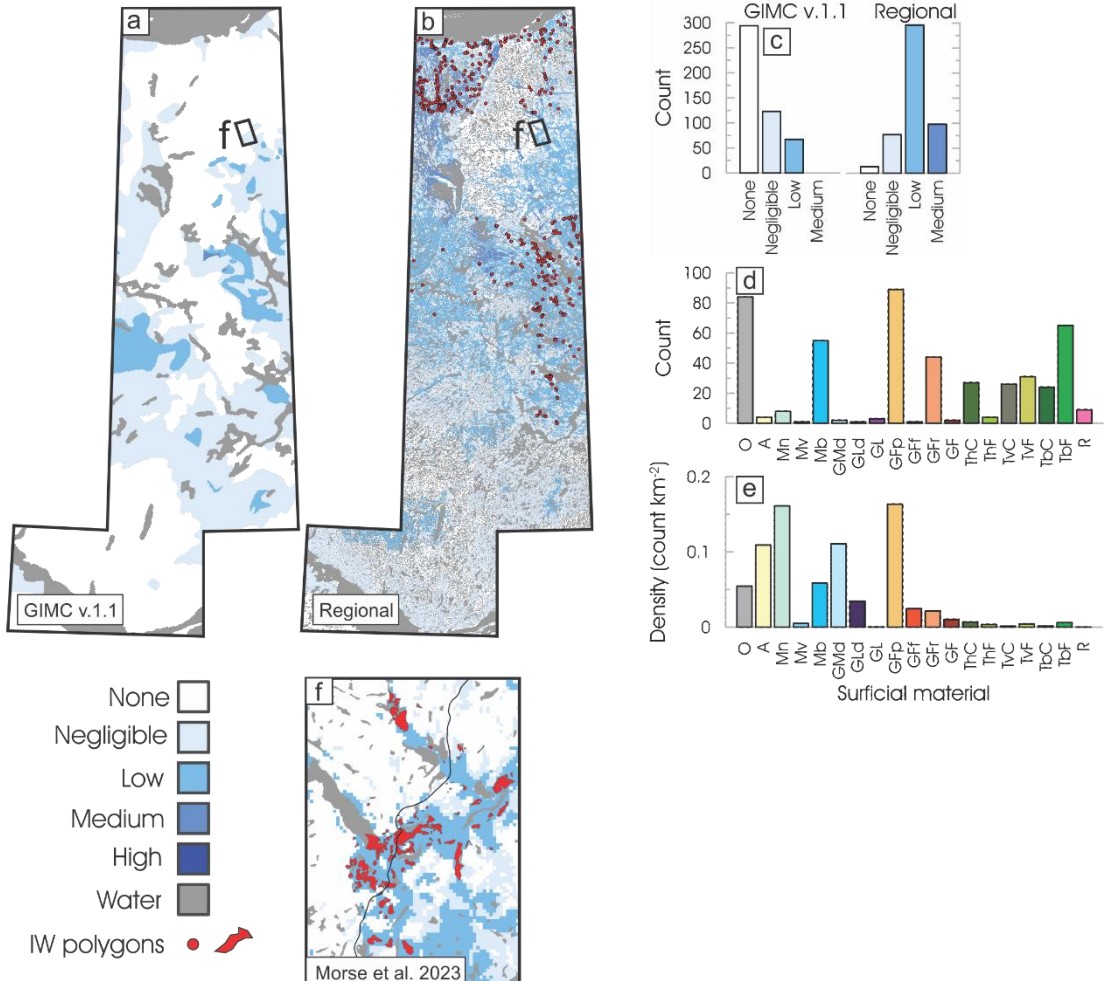

**Figure 7. Modelled wedge ice abundance from a) GIMC v.1.1, b) the regional modelling, with ice wedge polygons mapped on five Canadian Geoscience Maps overlain (n=484), c) histogram of mapped ice wedge polygon points from b) and corresponding ground ice abundance on the regional and national-scale mapping, respectively, d) histogram of surficial geology from the regional compilation underlying each mapped ice wedge polygon point, e) corresponding density of ice wedge polygon points in each surficial material unit; the area for each unit is based on coverage for the entire modelled domain, and f) ice-wedge polygons mapped along the proposed transportation infrastructure corridor (Morse et al., 2023).**

Though many ice wedge polygons were mapped in areas of till blanket on the CGM mapping (Fig. 7d) and by Morse et al. (2023), the percent coverage in this unit is very low (Fig. 7e). This is likely due to the widespread distribution of mudboils on till units and solifluction lobes on till slopes (Dredge et al., 1999; Kerr et al., 1996), which can obscure the surface expression of ice-wedge polygon troughs and ridges (Mackay, 1990). Polygons are observed in organic-capped depressions within till where mudboils and solifluction are absent, and ice wedges are observed in most flat outwash deposits, and glaciofluvial deposits such as eskers (Kerr et al., 1996; Wolfe et al., 2017). These observations and suitable climatic conditions suggest that all till units in the northern portion of the study region are likely subject to thermal contraction cracking and the accumulation





of vein/wedge ice in upper permafrost despite the lack of polygonal surface expression. This highlights a limitation of using imagery for validation of wedge ice modelling in surficial units with other active surface processes. In summary, the improved
representation of unconsolidated sediments on the regional compilation has increased modelling accuracy for wedge ice compared to the GIMC. Nevertheless, investigations into wedge ice volumes are required for further assessment of the validity of specific modelled abundance classes.

## 5.2 Implications

### 5.2.1 Infrastructure corridor assessment

Regional-scale ground ice modelling offers more useful information for initial cost or risk assessments on the effects of permafrost thaw on infrastructure than small-scale products (Fig. 8). Recent, broad-scale assessments have relied on the IPA map (Hjort et al., 2018; Streletskiy et al., 2023) or GIMC (Clark et al., 2022) as inputs. While these may be suitable for first-pass, aggregate estimates, the differences in ground ice conditions vary among modelling products (Fig. 8). The GIMC significantly underestimates the occurrence and abundance of ice-rich sediments along the proposed Yellowknife-Grays Bay
corridor for the reasons discussed above. The IPA map, based on the Permafrost map of Canada (Heginbottom et al., 1995), indicates the highest (medium) ice contents along the southern portion of the corridor, in contrast with the regional modelling and GIMC (Fig. 8). The variation in ground ice conditions north of Lac de Gras, from none to very high in the regional modelling, is not represented at all on the IPA map. Overall, the IPA map depicts the highest average ice content along the length of the corridor (low to medium), the GIMC the lowest (none to negligible), and the regional modelling between these
small-scale products (negligible to low). The IPA map may therefore offer more conservative (higher) total estimates of thaw risks or costs relating to infrastructure than the GIMC in areas of the Canadian Shield where regional-scale modelling is not yet available. However, the regional modelling captures far greater heterogeneity and thus offers more location-specific information useful for identifying specific thaw-susceptible sections of the corridor, which is critical for infrastructure planning and management, but not possible with the IPA map.





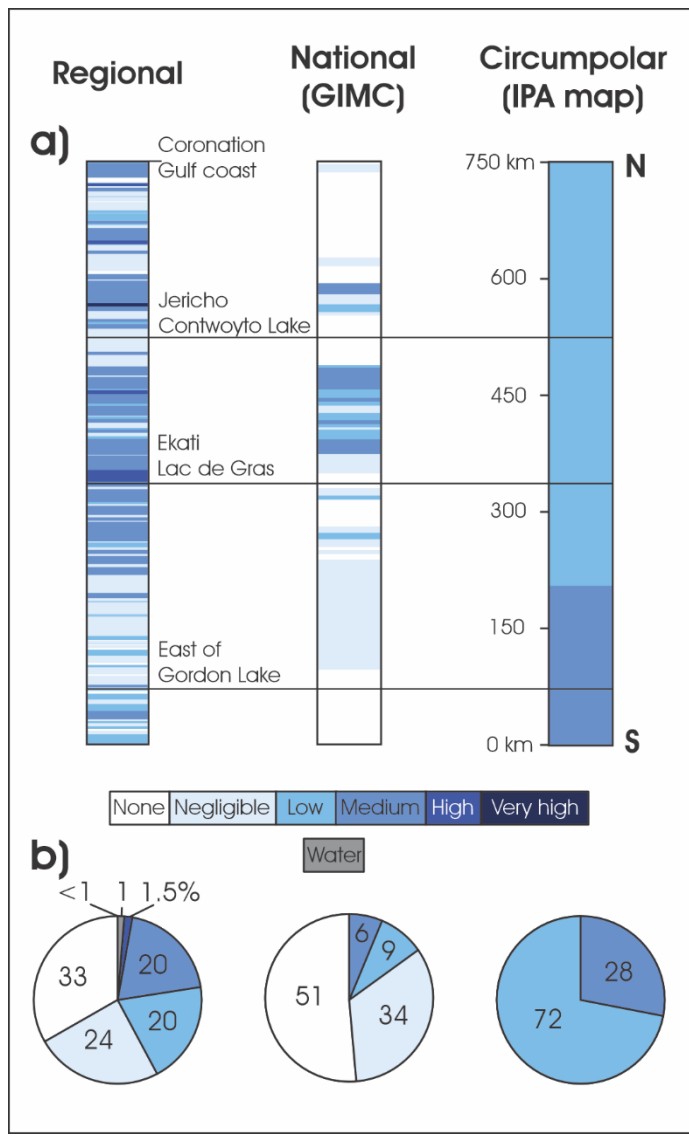

**Figure 8. a) Modelled ground ice abundance underlying the proposed Yellowknife-Grays Bay transportation infrastructure corridor route for the regional modelling, Ground ice map of Canada (GIMC), and IPA map, and b) the % of the length of the route represented by each ice abundance class.**

## 6 Conclusions

The results from this exercise examining the influence of surficial geology mapping scale on modelled ground ice abundance highlight that the occurrence of ice-rich terrain is significantly lower on the GIMC compared to the regional modelling over Canadian Shield terrain due to the underrepresentation, or complete absence of some unconsolidated sediment types on the national-scale surficial materials dataset. Available empirical datasets and observations indicate improved accuracy of the

regional-scale modelling. The inaccuracy in the GIMC, compared to the RC, likely occurs across much of the Canadian Shield,
where pockets of frost-susceptible unconsolidated sediments are prevalent among areas dominantly consisting of bedrock or
till veneer. Therefore, assessments based on the GIMC will significantly underestimate regional thaw risks on the Canadian
Shield. In areas with more homogenous cover, such as the western Arctic where thick and continuous frost-susceptible deposits
occur, the discrepancy in modelled abundance between scales is anticipated to be less pronounced. The improved
representation of heterogeneity in regional-scale ground ice modelling outputs are more useful than small-scale mapping for
reconnaissance-level assessments of ground ice along linear infrastructure routing, and to guide more detailed investigations.

## Data availability

All data used in the modelling and analysis are publicly available and cited in text.

## Author contributions

SAW conceived the original modelling approach. SAW, HBO, CD, developed the modelling routines. HBO and SAW
designed the analysis and prepared the paper. CD and HBO produced cartographic outputs; HBO produced the figures. RJHP
conducted data analysis and automated modelling routines. All authors contributed to edits of the paper prior to submission.

## Competing interests

The contact author has declared that none of the authors has any competing interests.

## Acknowledgements

This paper is NRCan contribution number 20230250. The research was supported by the Geological Survey of Canada's
Climate Change Geoscience Program and Geo-Mapping for Energy and Minerals (GEM-GeoNorth) Program. This paper is
NRCan contribution #20230250. We are grateful to S.L. Smith, S.V. Kokelj, and C.R. Burn for their constructive comments
that led to improvement of the manuscript prior to submission.

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
