# Peer review of "Effect of surficial geology mapping scale on modelled ground ice in Canadian Shield terrain"

_EGUsphere, 2024_

## Author Comment (AC1)

We thank both reviewers for taking the time to read the manuscript and for their helpful comments.

**Responses to reviewer 1. (marked in yellow).**

The aim of this manuscript is to compare the results of a ground ice model using surficial geology input layers of greatly differing scales and to compare the modelled ground ice content to empirical datasets. The study area, located within the Slave Geological Province of the Canadian Shield has an abundance of exposed bedrock and thin till. The distribution of these materials results in the underrepresentation of minority component materials in national-scale mapping (Fulton, 1995). Because bedrock is assumed to have no ground ice content and till veneers minimal, the GIMC (O'Neill et al, 2022) underrepresents ground ice compared to the regional-scale modelling presented in this manuscript that better reflects real ground conditions due to increased surficial geology mapping resolution which allows for delineation of more minority constituent materials which are comprised of unconsolidated sediments.

General Comments:

The manuscript is well written, easy to follow and contains figures that effectively illustrate the findings. I am very happy to see the method developed by O'Neill et al (2022) applied at a regional scale as this is when the utility and efficacy of the models can begin to be tested properly and begins to have more real-world benefits.

This manuscript primarily explores the idea of the impact of the spatial scale of input data on modelling results through the comparison of a model run with inputs at 1:5 000 000 and 1: 125 000 scale. That generalizations of the landscape based on map scale will underrepresent certain elements, which may be of significance, is not a new concept. While this concept is not new, it is not commonly explored in relation to ground ice. While the GIMC presents an interesting and novel approach to modelling ground ice distribution in Canada, there are limited use cases for a 1:5 000 000 scale product. It is unclear that comparison of the regional scale modelling to a national-scale model using a surficial geology layer developed specifically to look good on a wall map (line 169) is useful in particular; however the exploration of the concept is useful in general. Only a single geological region is examined within this manuscript rather than randomized sites distributed throughout the whole GIMC. The extreme change in scale of mapping means minority materials are better represented in the RC vs the GIMC. In the Slave Province, this results in increased modelled ground ice abundance due to increased representation of unconsolidated materials. I expect that in an area where bedrock is the minority material, e.g. in Northern Yukon, the expected difference between the RC and GIMC would be the opposite, less ground ice than predicted by the GIMC based on surficial geology. In this vein, I believe it would benefit this paper to include some more general statements on how the scale of model inputs are likely to affect the results in both discussion and conclusion sections.

Throughout the paper many different terms are used to describe mapping scale and many appear to be used interchangeably, please be consistent and use only as many terms as are necessary. E.g. small, broad, regional, national, circumpolar, hemispherical, finer.

We have changed instances of "broad-scale" to "small scale". We have kept "national-scale" as we indicate this is referring to 1:5 000 000. We removed instances of circumpolar and hemispherical in

reference to scale, but rather indicated "broad circumpolar regions" at the end of the first sentence of abstract.

On the figures showing the regional-scale modelling there appears to be subtle rectilinear boundaries which likely represent 125 000 map boundaries (e.g. Fig 2). This suggests some inconsistency in delineation or attribution of surficial geology units between map sheets. This limitation is worthy of discussion, is it something that could have been rectified during compilation?

Yes, this is due to the original surficial geology mapping and subtle differences in polygon delineation by individual mappers. This could not easily be rectified as it would require re-mapping of surficial geology polygons based on e.g., aerial photography or otherwise, which unfortunately was beyond the scope of the study.

Specific Comments:

13: hyphenate "regional-scale"

Implemented.

14: can you be more precise than "greater"

We considered adding reference to the % cover of unconsolidated sediments here, but decided this may be too detailed for the abstract and thus kept it unchanged.

15: remove "available"

Thanks, we have removed it.

32: the grainsize of the till is specified in general terms for thicker deposits but not for veneers

We have added that till veneers may be coarse grained.

41: see also McKillop et al, 2019. Predictive Mapping of the Variable Response of Permafrost Terrain to Climate Change for Optimal Roadway Routing, Design, and Maintenance Forecasting in Northern Canada

Thanks – missed this one and certainly relevant so we have added the reference.

49: hyphenate "national-scale"

Added.

58: remove the word "terrain"

Removed.

86: Undifferentiated units are allowable in the Deblonde et al Surficial Data Model however the mapping of methods of Olthof et al are perhaps unorthodox and so their definition of "undifferentiated" may not be consistent with Deblonde et al.

We have changed the wording slightly to clarify that this specific unit did not conform.

90: It would also be useful to note the number of surficial material classes represented in the study area by the GIMC.

We have added that there are 8 in the study area.

132: Specify the location within NT, Yellowknife?

We have indicated that it begins near Tibbett Lake.

149: instead of organic "terrain" use "materials" or "accumulations"

We have changed terrain to "materials"

Table 1. The use of italics vs border thicknesses should be reexamined here. Perhaps move water to the top and have the sum of uncondsolidated materials (this can include organics) shown below a thicker border.

We have implemented this change.

196: higher "modelled" ground ice abundance

Added.

198: suggest change "which is associated with" to "which results in"

Yes, we have changed it

229: re: incorporating line features. These line features do not have associated polygons because they cannot be delineated as such at the map scale. Discussion of including them is akin to saying that incorporating surficial geology data at a finer spatial scale would better represent the landscape and result in improved modelling. This is true and the main point of your paper but not made clear here.

This is true, but incorporating finer-scale surficial geology is not possible over much of the area as it may not exist or be compiled. We have added a sentence: "otherwise, polygons representing these features in larger-scale surficial mapping could be incorporated directly, but this is not widely available over broad regions".

293: "predicted" ground ice conditions

Added.

294: Slight distinction, the GIMC does not estimate the occurrence or abundance of sediments, but the ice content within the sediments.

True – we have changed to "ground ice"

314: Inaccuracy of the GIMC relative to RC will occur everywhere where terrain heterogeneity exceeds what can be represented a 1:5 000 000 scale. The direction of the effect will depend on the specific material present with the bias favouring whatever materials are dominant in the region.

This is a good point to emphasize, so we added "Inaccuracy of the GIMC can be expected elsewhere where the heterogeneity of surficial material exceeds what can be depicted at the 1:5,000,000 mapping scale."

317: The above comment applies to this line too.

320: Perhaps it would be a good idea to call attention to the need for detailed surficial geology mapping to support our understanding ground ice distribution at scales necessary for northern development.

Good idea. We have added this at the end.

---

## Author Comment (AC2)

We thank both reviewers for taking the time to read the manuscript and for their helpful comments.

**Responses to reviewer 2 (marked in yellow).**

This paper aims at comparing ground ice abundance modelling output from different scale surficial geology products for a region of the Canadian Shield. It uses an existing modelling method used for creating the GIMC, which was proven to underestimate ground ice abundance. Difference between two surficial geology scales are presented and validated using ancillary data, and implications of accuracy of ground ice abundance modelling are discussed for the Canadian Shield and along a proposed infrastructure route.

General comments

This paper presents novel and valuable insights into permafrost ground ice abundance modelling and mapping. It highlights the importance of scale and landscape heterogeneity, which is a very relevant challenge/issue related to products such as the GIMC and IPA map that needed to be addressed. The purpose is clear and is effectively reached using adequate methods. The conclusions are well supported by the results presented.

Overall, this paper is well-written and concise, but I think a few sections could benefit from additional information (discussed below).

Specific comments

The introduction is a little bit short in introducing the subject of ground ice and ground ice modelling. First paragraph could contain more information on the influence of surficial geology on ground ice abundance; why are we using surficial geology as an input to ground ice abundance modelling? Could also refer to existing datasets and why it is important to improve them, or even why it is important to quantify ground ice abundance.

The subject of ground ice modelling is described at length in our 2019 paper in the Cryosphere, and elsewhere, so we did not feel the need to repeat this introduction at length here, but rather cite relevant papers. We have added one sentence to establishing the link between surficial geology and frost susceptibility along with references that further deal with controls on ground ice abundance. We feel the first introductory sentences establish why ground ice is important: "Ground ice is a critical component of permafrost terrain and provides geotechnical strength to frozen ground. However, climate change is causing permafrost thaw and ground ice melt (Smith et al., 2022), resulting in widespread terrain subsidence (O'Neill et al., 2023), hillslope failure (Lewkowicz and Way, 2019), changes to hydrologic conditions (Walvoord and Kurylyk, 2016), and damage to infrastructure (Doré et al., 2016)."

Also, at line 35, there is a jump from geology of the study area to modelling methodology. I suggest adding "used for modelling ground ice abundance" after "methodology" and before "was" in sentence "modelling methodology was developed by O'Neill et al. (2019)…".

We have made this change, and added a paragraph break to separate the geology and modelling methodology.

In the Study Area section, some locations are mentioned, but are not presented on Figure 1, which makes it harder to understand the geological and climatic context. Suggest adding locations of the Great Slave Lowlands and Lac de Gras.

Lac de Gras is indicated in Figure 2. We decided to add the locations mentioned in text to Figure 2 instead of Figure 1 so that they could be examined in relation to ground ice conditions. Labeling on Figure 1 is also challenging as the permafrost zones are already indicated, as is Yellowknife, so the Figure would become cluttered. We have added a label of GSL to show Great Slave Lowlands in Figure 2.

In the Methods section, the different products used become obscured. I suggest reviewing and keeping a constant terminology for each product/group to avoid confusion:

Line 84: What are these ten CGMs? Are they the RC surficial geology maps mentioned at line 80?

The first line of Methods section indicates that the 11 maps comprise the regional compilation.

Line 93: What is the meant by "at the national scale"? Is that the product used to generate the GIMC? What product is that?

Earlier on line 83 we indicate "As with the 1:5 000 000 scale national surficial compilation" which introduces the national-scale product. We have clarified the wording to indicate "national-scale surficial compilation".

Line 90: "surficial material classes" are mentioned, but the term "units" is used at line 92-93. This occurs in other places throughout the text.

We have changed all instances associated with surficial geology to "units" while ground ice abundances are referred to as "classes".

Line 99: What is meant by the "other model"?

We have clarified the wording here to "other data layers used in the model"

In the Results section, there is mention of "unconsolidated sediments and organic terrain associated with ground ice" (Line 149 and Table 1). Again, I think information on what makes certain types of surficial deposits susceptible to being more ice-rich than others is lacking. This could be addressed in the introduction, as mentioned above.

As indicated above, we have now linked surficial geology to frost susceptibility in the introduction. We believe those requiring further information can consult the references provided.

Technical corrections

I somewhat question the sectioning of the Results and Discussion sections:

Results of the validation (Section 5.1) and infrastructure corridor assessment (Section 5.2.1) belong in the results section.

I suggest keeping the discussion section for the implications of the results only (i.e., impact of homogeneity/heterogeneity of deposits, inclusion of linear features, model exceptions for ice-marginal deposits, limitations of wedge ice modelling based on imagery, etc.)

These implications could also benefit from being further discussed, including within are broader context (E.g., impact of homogeneity/heterogeneity of deposits in other regions/publications, how can linear features be included in such modelling exercises, etc.)

Thanks for this suggestion on formatting. However, we believe it is appropriate to structure the results as summarizing the surficial geology datasets and ground ice model outputs. The validation and infrastructure corridor assessment place these results and discuss them in the context of past work/observations, which is consistent with the typical scope of a discussion section. Since we, the editor, and Reviewer 1 did not believe the structure required adjustment we have kept as is. We have added a sentence in the conclusion about the broader context of the results, also in response to a comment from reviewer 1. We also added a sentence in the section on heterogeneity (l.221) on the broader context of the results for the GIMC: "In contrast, the GIMC may overestimate the distribution of ground ice abundance in areas where frost-susceptible deposits are dominant and where smaller bedrock outcrops or areas of till veneer are not represented on the surficial mapping. "

---

## Referee Report (RR1)

Review of revised manuscript egusphere-2024-68

The authors have adequately implemented comments in the revised manuscript, and I accept their justification for not making some of the proposed changes. My recommendation is that the manuscript can now be published without further revisions.